# MF-LAL: Drug Compound Generation Using Multi-Fidelity Latent Space Active Learning

**Peter Eckmann** [1]  **Dongxia Wu** [1]  **Germano Heinzelmann** [2]  **Michael K. Gilson** [3][4]  **Rose Yu** [1]

## Abstract

Current generative models for drug discovery primarily use molecular docking as an oracle to guide the generation of active compounds. However, such models are often not useful in practice because even compounds with high docking scores do not consistently show real-world experimental activity. More accurate methods for activity prediction exist, such as molecular dynamics based binding free energy calculations, but they are too computationally expensive to use in a generative model. To address this challenge, we propose Multi-Fidelity Latent space Active Learning (MF-LAL), a generative modeling framework that integrates a set of oracles with varying cost-accuracy tradeoffs. Using active learning, we train a surrogate model for each oracle and use these surrogates to guide generation of compounds with high predicted activity. Unlike previous approaches that separately learn the surrogate model and generative model, MF-LAL combines the generative and multi-fidelity surrogate models into a single framework, allowing for more accurate activity prediction and higher quality samples. Our experiments on two disease-relevant proteins show that MF-LAL produces compounds with significantly better binding free energy scores than other single and multi-fidelity approaches ($\sim$ 50% improvement in mean binding free energy score). The code is available at https://github.com/Rose-STL-Lab/MF-LAL.

[1]Department of Computer Science and Engineering, UC San Diego, La Jolla, California, United States [2]Departamento de Física, Universidade Federal de Santa Catarina, Brazil [3]Department of Chemistry and Biochemistry, UC San Diego, La Jolla, California, United States [4]Skaggs School of Pharmacy and Pharmaceutical Sciences, UC San Diego, La Jolla, California, United States. Correspondence to: Peter Eckmann <peckmann@ucsd.edu>, Michael K. Gilson <mgilson@health.ucsd.edu>, Rose Yu <roseyu@ucsd.edu>.

*Proceedings of the 42$^{nd}$ International Conference on Machine Learning*, Vancouver, Canada. PMLR 267, 2025. Copyright 2025 by the author(s).

## 1. Introduction

Generative models for *de novo* drug design have gained significant interest in machine learning for their promised ability to quickly generate new compounds for specific applications. However, generating compounds with real-world biological activity remains a fundamental challenge (Handa et al., 2023; Coley et al., 2020). One of the main difficulties is the computational evaluation of compound-protein binding affinities. The generated compounds are often highly novel, so an activity predictor trained with existing experimental data is insufficient due to poor out-of-distribution generalization (Chatterjee et al., 2023; Ji et al., 2022). Instead, physics-based methods that model 3D interactions between compound and target are commonly used.

Due to its speed, molecular docking is the prevalent physics-based method to evaluate novel compounds by generative models (Eckmann et al., 2022; Jeon & Kim, 2020; Lee et al., 2023; Noh et al., 2022; Fu et al., 2022; Peng et al., 2022; Guan et al., 2023a;b). However, docking is a relatively poor predictor of activity (Pinzi & Rastelli, 2019; Handa et al., 2023; Coley et al., 2020; Feng et al., 2022), so it would be desirable to apply more accurate binding free energy calculation techniques (Pinzi & Rastelli, 2019; Feng et al., 2022). These techniques, based on molecular dynamics simulations, are currently considered the most reliable approach to predict affinity (Moore et al., 2023; Cournia et al., 2021). However, they have not been used by generative models due to their high computational cost (Thomas et al., 2023), with a single compound-protein pair taking hours to days to simulate on a powerful computer (Wan et al., 2020). Thus, neither docking nor binding free energy techniques alone can guide the real-world application of generative models.

Multi-fidelity surrogate models aim to fuse multiple data sources as oracles spanning a range of accuracy and cost (Fernández-Godino, 2023).They are frequently learned using an active learning approach, where a model selects or generates queries that it is most uncertain about to send to a chosen oracle (Ren et al., 2021). The results from the oracle are then added to the training data of the model. We will focus on "query synthesis" approaches (Angluin, 1988), where the model generates its own queries to send to the oracles, speeding up learning compared to approaches that

query oracles with samples from a fixed candidate set (Guo et al., 2021; Zhu & Bento, 2017; Morand et al., 2022).

Combining docking (low fidelity) and binding free energy (high fidelity) using multi-fidelity surrogate models holds promise to make generative models more practical. Yet, the use of multi-fidelity methods in drug discovery has been limited. Prior work from (Hernandez-Garcia et al., 2023) uses a generative model to generate query compounds with high acquisition function values computed by a *separate* multi-fidelity surrogate model. However, since we want to generate query compounds to send to oracles at multiple fidelity levels, the distribution of optimal query compounds may differ across fidelities. A separate generative model is not aware of such differences across fidelity levels, hence it cannot send queries to the multi-fidelity oracles efficiently.

We aim to address the problem of multi-fidelity generation with Multi-Fidelity Latent space Active Learning (MF-LAL), an integrated framework for compound generation using multi-fidelity active learning. Instead of separating the generative model and surrogate model, we perform surrogate modeling and generation together at each fidelity level using a sequence of hierarchical latent spaces. This improves the quality of generated queries because there is a separate latent space and decoder specialized for each fidelity, and improves surrogate modeling because each latent space can be organized for predicting at just that level. Information is shared between fidelity levels using networks that map from lower to higher fidelity latent spaces. We use both docking and binding free energy methods as oracles in our multi-fidelity environment to achieve a favorable trade-off between cost and accuracy. In summary,

- We introduce a novel multi-fidelity generative modeling framework, **MF-LAL**, which integrates data from multiple fidelity levels to generate high-quality samples at the highest fidelity (binding free energy).

- We employ an active learning approach with a novel query generation technique that ensures compounds generated at higher fidelities also scored well at lower fidelities, improving the quality of generated samples.

- We evaluate MF-LAL and state-of-the-art baseline methods on a real-world task of optimizing the binding free energy of compounds against two disease-relevant human proteins, and find that MF-LAL generates compounds with significantly better scores than baselines ($\sim 50\%$ improvement in mean binding free energy).

## 2. Related Work

### 2.1. Molecular generative models

Generative models in drug discovery have gained much interest for their ability to quickly generate compounds with desired properties (Paul et al., 2021). Early works (Jin et al., 2018; Gómez-Bombarelli et al., 2018; You et al., 2018) focus on properties such as the octanol-water partition coefficient (logP) or quantitative estimate of drug-likeness (QED), which are of very limited practical utility (Coley et al., 2020; Xie et al., 2021). More recently, there has been an understanding that the binding affinity to a targeted protein is much more relevant for practical drug discovery (Xie et al., 2021; Eckmann et al., 2022; Fu et al., 2022).

One approach to guide generative models in optimizing compound binding affinity is to use an oracle for compound evaluation. This oracle can be applied to reinforcement learning (Jeon & Kim, 2020; Fu et al., 2022; Mazuz et al., 2023), VAEs (Eckmann et al., 2022; Noh et al., 2022), genetic algorithms (Spiegel & Durrant, 2020; Fu et al., 2022), diffusion models (Lee et al., 2023; Hoogeboom et al., 2022; Wu et al., 2024; Zhou et al., 2024), or other generative frameworks (Zhu et al., 2024). All of them use docking software, such as AutoDock (Morris et al., 2009), as the oracle, because it is the only reasonably fast option. However, docking is known to be inaccurate (Pinzi & Rastelli, 2019), and compounds with high docking scores do not consistently show experimental activity (Handa et al., 2023; Coley et al., 2020; Feng et al., 2022).

Molecular dynamics-based binding free energy calculations are much more accurate than docking (Moore et al., 2023; Cournia et al., 2021), but have not yet been applied to *de novo* generative drug design due to their high computational cost (Thomas et al., 2023). While Ghanakota et al. (2020) use binding free energy calculations in combination with a molecular generative model, they focus on the optimization of a known lead compound. This allows them to rely on much cheaper relative binding free energy calculations, as opposed to the absolute binding free energy (ABFE) calculations needed for *de novo* design (Cournia et al., 2017).

Structure-based generative models are trained on 3D structures of protein-ligand pairs, and aim to predict a 3D ligand that fits in a given protein pocket with high binding affinity. Techniques include autoregressive generation (Peng et al., 2022) and diffusion modeling (Guan et al., 2023a;b). Despite not needing an oracle like docking during the generation process, the generated compounds are still evaluated with docking as a post-processing step. This means structure-based generative models do not avoid the issue of inaccurate binding affinity prediction.

### 2.2. Multi-fidelity surrogate modeling

Multi-fidelity modeling methods aim to fuse multiple data sources of variable accuracy and cost (Fernández-Godino, 2023), and are widely used in scientific fields for surrogate modeling and uncertainty quantification (Brevault et al., 2020). A popular choice of surrogate model is a Gaussian

process (GP), which performs well in low data settings and produces well-calibrated uncertainty estimates (Brevault et al., 2020). One such technique to apply GPs to multi-fidelity modeling is described by Wu et al. (2020), where a downsampling kernel is used to output predictions at each fidelity level. Other multi-fidelity surrogate modeling approaches utilize neural processes (Yating & Lin, 2020; Wu et al., 2022; 2023; Niu et al., 2024) and ordinary differential equations (Li et al., 2022b) as an alternative to GPs.

Multi-fidelity models are frequently trained in an active learning fashion, where one uses an estimate of a model's uncertainty to most efficiently acquire more datapoints from an oracle (Ren et al., 2021). In the multi-fidelity setting, this means iteratively querying across both the sampling space and each different fidelity oracle (Li et al., 2022a; Hernandez-Garcia et al., 2023). Traditional active learning involves selecting from a fixed candidate set with the highest acquisition function value to query oracles with, which limits the training set to only existing samples. It also limits how much the model can learn with each query, since the maximally informative sample may not be present in the candidate set. Query synthesis approaches (Angluin, 1988) have been proposed to avoid this problem by using a generative model to generate new queries. Hernandez-Garcia et al. (2023) have applied these ideas to drug discovery problems by training a generative model to optimize the acquisition function computed by a separate multi-fidelity surrogate model, which does not take into account the different distributions of optimal query compounds at each fidelity.

## 3. MF-LAL

We introduce Multi-Fidelity Latent space Active Learning (MF-LAL), an integrated framework for compound generation using multi-fidelity Bayesian active learning. An overview of our framework is shown in Figure 1. We encode molecules into a hierarchy of latent spaces (left panel), one for each fidelity, and learn surrogates that predict the oracle output based on the latent vectors (middle panel). These oracles are used to reverse optimize in the latent spaces to generate new compounds with high predicted scores. The generated molecules are fed to an oracle at a chosen fidelity in an active learning loop, the output of which is used to re-train the latent representation and surrogate models (right panel). After training, we use the surrogates to reverse optimize in the highest fidelity latent space and generate compounds with high property scores at the highest fidelity. See Figure 2 for a diagram of the network architecture, and Appendix A for further details of the model.

### 3.1. Learning multi-fidelity latent representations

**Problem setup.** A multi-fidelity environment consists of a set of oracles $\{f_1, \ldots, f_k, \ldots, f_K\}$ that predict a

property of interest, where the accuracy and cost of the predictions increase with the fidelity level $k$. We have a multi-fidelity dataset $\mathcal{D}$ consisting of $K$ distinct sets of molecule-activity pairs, one for each fidelity, $\mathcal{D} = \{\{x_1^{(i)}, y_1^{(i)}\}_{i=1}^{N_1}, \ldots, \{x_K^{(i)}, y_K^{(i)}\}_{i=1}^{N_K}\}$. Each $y_k^{(i)} = f_k(x_k^{(i)})$ is the result from querying oracle $f_k$ with molecule $x_k^{(i)}$. The molecules are drawn from unknown distributions $p_1^*, \ldots, p_K^*$. We aim to approximate these distributions using generative models $p_{\theta_1}, \ldots, p_{\theta_K}$ with parameters $\theta_1, \ldots, \theta_K$. Note that $p_1^* \neq \ldots \neq p_K^*$, meaning we must learn separate generative models for each fidelity level, as opposed to previous approaches that learn a single generative model for all fidelities.

**Latent representation.** To learn the generative models, we first learn an encoding of the input molecule to the lowest fidelity latent space. Specifically, we use a single probabilistic encoder $q_\phi$ parameterized by $\phi$ that encodes a molecule $x$ into mean and variance vectors $\mu_1$ and $\sigma_1$. The latent vector $\mathbf{z}_1 \sim \mathcal{N}(\mu_1, \sigma_1)$, corresponding to the first (lowest) fidelity, is sampled from the resulting distribution. Since we want a separate latent space at each fidelity level, we define a set of probabilistic networks $h_{\xi_1}(\mathbf{z}_1), \ldots, h_{\xi_{K-1}}(\mathbf{z}_{K-1})$ with parameters $\xi_1, \ldots, \xi_{K-1}$ that pass information between latent spaces. Specifically, $h_{\xi_k}$ takes the vector $\mathbf{z}_k$ as input and outputs a mean and variance vector in the subsequent latent space, $\mu_{k+1}, \sigma_{k+1}$. We sample from this distribution to obtain latent vector $\mathbf{z}_{k+1}$, i.e. $\mathbf{z}_{k+1} \sim \mathcal{N}(\mu_{k+1}, \sigma_{k+1})$. We also define a set of probabilistic decoder networks $p_{\theta_1}(\cdot|\mathbf{z}_1), \ldots, p_{\theta_K}(\cdot|\mathbf{z}_K)$ to reconstruct the original molecule $x$ from the latent vectors. The use of a specialized decoder for each fidelity level improves reconstruction quality compared to previous methods that only use one, thus making the generated samples more tailored for their fidelity level.

We represent molecules using SELFIES strings (Krenn et al., 2020). The encoder and decoder of MF-LAL are fully-connected neural networks that use a flattened, one-hot encoded SELFIES string. See Table 3 for comparison with other encoder and decoder designs.

**Surrogate modeling.** In order to generate molecules with high property scores, we aim to learn differentiable surrogates $\hat{f}_1, \ldots, \hat{f}_K$ that approximate the oracles and use them for reverse optimization in the latent spaces. Each surrogate $\hat{f}_k$ maps from its corresponding latent vector $\mathbf{z}_k$ to an estimate of $f_k$. We use gradient-based optimization to find a point in a given latent space that has a high property score predicted by the surrogate ("reverse optimization"), which can then be decoded to a molecule (Gómez-Bombarelli et al., 2018). The $h_{\xi_k}$ networks, which pass information between latent spaces, allow us to re-use information learned about the molecule's binding properties at lower fidelities to aid

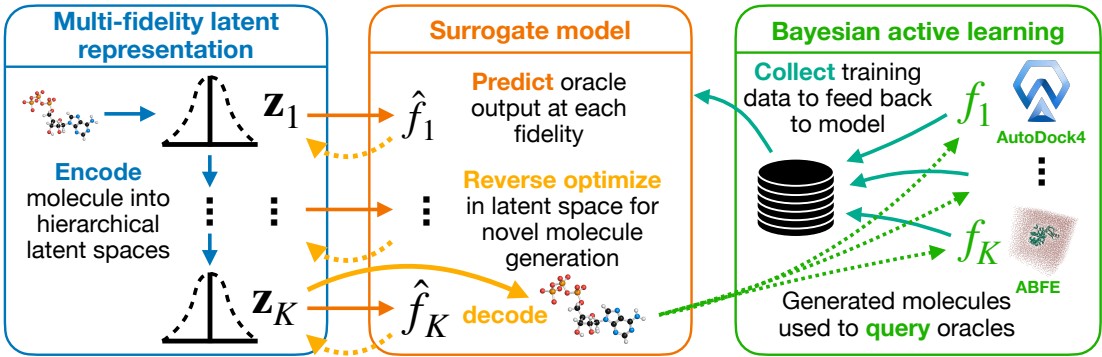

*Figure 1.* **Overview of Multi-Fidelity Latent space Active Learning (MF-LAL).** Left: molecules are encoded into a hierarchy of latent spaces. Middle: surrogate models predict the oracle outputs based on the latent vectors, and reverse optimization is performed in the latent spaces to generate high scoring compounds. Right: generated molecules are sent to the oracles and the results are used to re-train the latent representation and surrogate models.

in prediction at the higher fidelities without having to re-learn it using large amounts of high-fidelity data. This is because training the surrogate models organize each latent space (Tevosyan et al., 2022) for property prediction at that level, and so the latent vectors contain information about the binding properties useful for predicting the oracle output that can then be passed to higher fidelities. Additionally, the use of separate latent spaces for each fidelity level, as opposed to previous approaches that use only a single latent space shared across all levels, improves surrogate modeling performance because each latent space can be organized for prediction at just that level.

We use stochastic variational Gaussian process models (SVGPs, (Hensman et al., 2015)) with hyperparameters $\lambda_1, \ldots, \lambda_K$ (which define the mean and covariance kernels of the GP) for the surrogates. We chose SVGPs for their speed of training and ability to train with minibatches. Specifically, $\hat{f}_k$ is given by $\hat{f}_k \sim \mathcal{GP}(m_{\lambda_k}(x), \Sigma_{\lambda_k}(x, x'))$ where $m(x)$ and $\Sigma(x, x')$ are the mean and covariance kernels of the SVGP. To train the model, we jointly minimize the evidence lower bound (ELBO) (Kingma & Welling, 2014) of the latent encodings and marginal log likelihood (MLL) of the GP models (see Appendix A.1 for the full loss equation). While the loss is only evaluated at fidelity $k$, it is backpropagated through to all lower fidelities. Additionally, in our implementation, we approximate the MLL GP loss using the ELBO (Hensman et al., 2015) for improved scalability.

### 3.2. Bayesian active learning for sample-efficient training

Training a multi-fidelity surrogate model requires significant computational resources, especially to gather data at the highest fidelity level. Instead of passively collecting training data, we develop a Bayesian active learning approach to

**Algorithm 1** Active learning for MF-LAL

**Require:** a multi-fidelity dataset $\mathcal{D}$ consisting of a set of initial training examples, number of compounds to generate to generate at lower fidelities $M$
1:   $k \leftarrow 1$
2:   **while** computational budget is not exceeded **do**
3:      train model on data $\mathcal{D}$ (Eq. 2)
4:      $x \leftarrow$ generateHighScoringMols($k$, $M$, 1) (Algorithm 2)
5:      query $f_k(x)$ and save result in $y$
6:      $\mathcal{D}_k \leftarrow \mathcal{D}_k \cup \{(x, y)\}$
7:      **if** $k < K$ **and** $\Sigma_{\lambda_k}(\mathbf{z}_k) < \gamma_k$ **then**
8:         $k \leftarrow k + 1$
9:      **end if**
10: **end while**

efficiently query the oracles, allowing us to make fewer queries to the most expensive oracles. As show in Algorithm 1, our active learning cycle consists of first generating a molecule to query at a chosen fidelity, querying the oracle to obtain the property score, appending the result to the dataset, and then retraining the model. We repeat the process until some computational budget is reached.

Similar to (Kandasamy et al., 2016), we start with only querying the oracle at the lowest fidelity level $k = 1$ and increase to higher fidelities when the model's uncertainty falls below certain thresholds. We use the posterior variance, $\Sigma_{\lambda_k}$, of the GP surrogate $\hat{f}_k$ to measure the model's uncertainty. Specifically, during active learning, we repeatedly generate a latent vector $\mathbf{z}_k$ at fidelity $k$ that decodes to a query compound. If $\Sigma_{\lambda_k}(\mathbf{z}_k) < \gamma_k$, where $\gamma_k$ is the uncertainty threshold (which we treat as a hyperparameter), then we permanently increment $k$ by one for all subsequent queries. Otherwise, $k$ remains the same. Once at the highest fidelity, we keep running active learning until some compu-

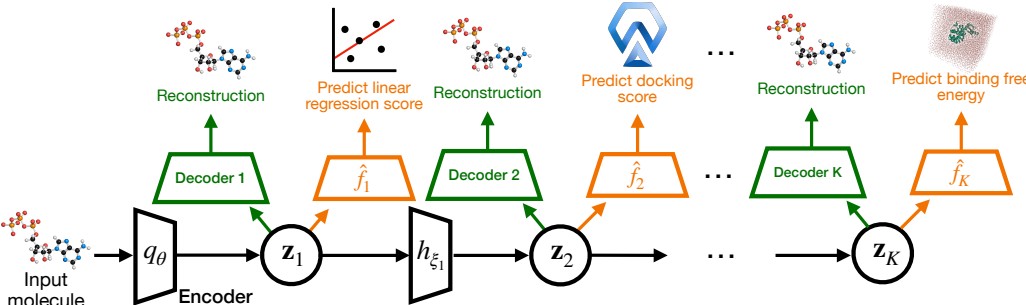

*Figure 2.* **Diagram of MF-LAL architecture**. An input molecule is encoded into the first latent space $\mathbf{z}_1$ using a neural network. The networks $h_{\xi_1}, \dots, h_{\xi_{K-1}}$ transform points in $\mathbf{z}_1$ to the higher fidelity latent spaces. Each latent space has an associated decoder, which reconstructs the original molecule, and a GP surrogate model for that fidelity level.

tational budget is reached. We use this stepwise approach to ensure all surrogates have enough training data to make accurate predictions in high property areas of the latent spaces, thus leading to high property compounds being generated.

We generate query molecules from the latent space using the upper confidence bound (Auer, 2002) as the acquisition function. The method we use to generate molecules is described later in Section 3.3. To ensure that generated compounds remain similar to the training set of drug-like molecules, we also add an L2 regularization term on the latent vector. The acquisition function is thus given by

$$a(\mathbf{z}_k^{(i)}, k) = m_{\lambda_k}(\mathbf{z}_k^{(i)}) + \beta \Sigma_{\lambda_k}(\mathbf{z}_k^{(i)}) - ||\mathbf{z}_k^{(i)}||_2^2 \quad (1)$$

where $\mathbf{z}_k^{(i)}$ is a point in latent space $k$ and $\beta$ is an exploration hyperparameter. $m$ and $\Sigma$ are the mean and covariance kernels of the GP surrogate. We set $\beta = 1$ during active learning, and $\beta = 0$ after during inference to only focus on the most promising compounds.

### 3.3. Generating molecules with high property scores

Our goal is to generate compounds at fidelity $k$ that maximize some generation objective. To accomplish this, we perform gradient-based optimization to find a point $\mathbf{z}_k^{(i)}$ in the $k$th latent space that maximizes the generation objective, and then decode $\mathbf{z}_k^{(i)}$ into a molecule using $p_{\theta_k}$. For our generation objective, we do not want to simply maximize $\hat{f}_k$, but instead the upper confidence bound (Eq. 1) to ensure exploration during active learning. In addition, we also introduce a novel likelihood-based term to the generation objective that encourages the model to only sample compounds at higher fidelities that also scored well at the lower fidelities. Specifically, when generating a molecule at fidelity $k$, we maximize the likelihood that the molecule would also be generated at fidelity $k-1$ with a high property score. This additional term greatly restricts the area

of the chemical space explored by the high fidelity oracles, reducing the computational cost wasted on non-promising areas and making the use of high-cost oracles feasible. It also means the higher fidelity latent spaces encode a more limited distribution of compounds, improving the quality of samples generated from those latent spaces. Indeed, we show that the likelihood term is critical for strong performance (Table 3).

To compute the likelihood of a point $\mathbf{z}_k^{(i)}$ at fidelity $k$, we first generate a set of $M$ high-scoring compounds at fidelity $k-1$. Next, we map those points to a sum of Gaussians in the $k$th latent space using $h_{\xi_{k-1}}$, giving us a set of $M$ parameters $\{(\mu_k^{(j)}, \sigma_k^{(j)})\}_{j=1}^M$. We then measure the likelihood of point $\mathbf{z}_k$ in the generated sum of Gaussians distribution (see Appendix A.1 for mathematical details). This guarantees that the compounds generated in latent space $k$ are also likely to have been generated in $k-1$ with high scores. Thus, we effectively reduce the size of the chemical space that must be explored at fidelity $k$ to only compounds that have already shown promise at the lower fidelities.

The full generation algorithm is detailed in Algorithm 2. In our implementation, we vectorize optimization across all $M$ latent space points simultaneously. In order to encourage diversity in generated compounds, we also add a term to the generation objective that measures the average pairwise cosine similarity between the $M$ points (Appendix A.1)

## 4. Experiments

### 4.1. Experimental setup

We define a multi-fidelity environment for binding affinity which uses four oracles, each of which takes a molecule as input and outputs an estimate of its binding affinity with increasing accuracy:

1. **Linear regression** ($f_1$). Simple linear regres-

---

**Algorithm 2** MF-LAL molecule generation procedure

---

**Require:** fidelity $k$ to optimize compounds for, exploration/exploitation hyperparameter $\beta$, number of compounds to generate at lower fidelities $M$

1: **procedure** GENERATEHIGHSCORINGMOLS($k, M, \beta$)
2:     **procedure** GETTOPLATENTPOINTS($k, M, \beta$)
3:         **for** $i$ in $1..M$ **do**
4:             initialize $\mathbf{z}_k^{(i)} \sim \mathcal{N}(\mathbf{0}, \mathbf{I})$ as a starting point for gradient-based optimization
5:             **if** k == 1 **then**
6:                 find $\mathbf{z}_k^{(i)}$ that maximizes Eq. 1 via gradient descent
7:             **else**
8:                 $\mathbf{z}_{k-1}^{(1)}, \ldots, \mathbf{z}_{k-1}^{(M)} \leftarrow$ getTopLatentPoints($k - 1, M, \beta$)
9:                 **for** $j$ in $1..M$ **do**
10:                     $\mu_k^{(j)}, \sigma_k^{(j)} \leftarrow h_{\xi_{k-1}}(\mathbf{z}_{k-1}^{(j)})$
11:                 **end for**
12:                 find $\mathbf{z}_k^{(i)}$ that maximizes Eq. 1 + Eq. 3 via gradient descent
13:             **end if**
14:         **end for**
15:         **return** $\mathbf{z}_k^{(1)}, \ldots, \mathbf{z}_k^{(M)}$
16:     **end procedure**
17:     $\mathbf{z}_k^{(1)}, \ldots, \mathbf{z}_k^{(M)} \leftarrow$ getTopLatentPoints($k, M, \beta$)
18:     **for** $i$ in $1..M$ **do**
19:         $x^{(i)} \sim p_{\theta_k}(\cdot|\mathbf{z}_k^{(i)})$
20:     **end for**
21:     **return** $x^{(1)}, \ldots, x^{(M)}$ **if** $k < K$ **else** $x^{(1)}$   ▷ only need one compound at highest fidelity
22: **end procedure**

---

sion model trained on experimental data from the BRD4(2)/c-MET target from BindingDB (Liu et al., 2007) to predict the $K_i$, a measure of binding affinity. Morgan fingerprints are used to represent the molecule.

2. **AutoDock4** ($f_2$) (Morris et al., 2009). Uses 3D geometric and charge information from the protein and compound to estimate the binding energy.

3. **Ensembled AutoDock4** ($f_3$) (Morris et al., 2009). Same as above, except we dock the compound into the binding pockets of 8 BRD4/5 c-MET cocrystal structures that were solved with different known ligands, and then take the minimum predicted energy. This ensemble approach is generally more accurate than using a single protein structure (Amaro et al., 2018).

4. **Absolute binding free energy (ABFE)** ($f_4$) (Heinzelmann & Gilson, 2021). A binding free energy method applicable to *de novo* drug discovery that uses molecular dynamics simulations to accurately predict the binding energy.

We target the BRD4(2) and c-MET proteins (PDB 5UF0 and 5EOB), both of which are implicated in human cancer development, although through different biological mechanisms. We chose these targets because ABFE is already well-validated on them and known to have good agreement with experimental data (Heinzelmann & Gilson, 2021). See Appendix B for further experimental details and analysis of the oracles, including experiments confirming that our higher fidelity oracles are more costly yet more accurate at distinguishing experimental actives from inactives.

Each model is provided with an initial dataset of random ZINC250k (Irwin et al., 2012) compounds queried at each fidelity (see Appendix B for further details). To compare models, we run each in an active learning loop using a fixed computational budget of 7 days, and then generate 15 unique compounds at the highest fidelity predicted to have the best scores. We then run these compounds through ABFE and compare their scores. Based on the real-world use case of our method, and following other works (Lee et al., 2023; Luo et al., 2021; Fu et al., 2022), we focus particularly on the properties of the top 3 generated compounds. This is because drug campaigns would take only the top compounds generated and use them as starting points for further optimization, so the binding affinities of the top compounds are the most relevant for measuring performance. See Appendix C for additional results showing the oracle-predicted binding energy of the generated query compounds over the active learning process, and the reconstruction accuracy for each fidelity decoder during training.

### 4.2. Baselines

We compare MF-LAL with the following baselines:

- **SF-VAE (only ABFE / only docking)** (Gómez-Bombarelli et al., 2018). Uses a simple single-fidelity GP as an activity surrogate model that is used to guide optimization in the latent space of a vanilla VAE, representing a simple single-fidelity approach. This consists of two separate baselines, one where the GP is trained only on ABFE data and one where it is trained on only docking data.

- **REINVENT (only ABFE / only docking)** (Olivecrona et al., 2017). An RL-based molecular generation technique which we use to optimize ABFE, and separately, docking score. Represents a simple single-fidelity approach using a modern generative model.

- **RGA (only docking)** (Fu et al., 2022). A genetic algorithm-based method that uses a 3D neural network on both the protein and ligand to prioritize mutations and crossovers. The docking score is used as the reward function, making it a single-fidelity method.

- **VAE + 4x SF-GP**. Uses a vanilla VAE model for molecule generation and four independent GP surrogates for activity prediction, one for each fidelity, all using the single latent space as input. To be contrasted with MF-LAL, which uses multiple connected latent spaces instead of a single one.

- **VAE + MF-GP**. Similar to above, except using a single multi-fidelity GP model activity surrogate (Wu et al., 2020) instead of four independent single-fidelity GP models.

- **MF-AL-GFN** (Hernandez-Garcia et al., 2023) GFlowNet generative model used to optimize the predicted score from a multi-fidelity GP model. This baseline represents the state of the art in multi-fidelity generation, where generative and surrogate models are separated.

- **MF-AL-PPO** (Hernandez-Garcia et al., 2023). Same as above except using the PPO RL algorithm instead of a GFlowNet as the generative model.

- **MF-GP + ZINC250k**. Active learning using a multi-fidelity GP model (Wu et al., 2020) with a fixed candidate set consisting of ZINC250k compounds (Irwin et al., 2012). This baseline represents a non-query synthesis approach to multi-fidelity active learning.

- **Pocket2Mol** (Peng et al., 2022). 3D structure-based drug design model that takes a protein pocket as input and outputs a 3D molecule via diffusion. Unlike the other methods, does not use any binding affinity oracle during generation.

- **DecompDiff** (Guan et al., 2023b). Same type of model as above, but makes additional improvements to the generation process by decomposing generation into scaffold and motif stages.

- **TAGMol (only docking)** (Dorna et al., 2024). 3D structure-based diffusion model that conditions generation on the protein pocket. A property predictor is introduced to guide the diffusion process towards ligands that have high predicted affinity. In our case, the property predictor is a single-fidelity predictor of the docking score.

The first three baselines are single-fidelity methods, where we use both only ABFE and only docking as the single fidelity. The next five baselines, as well as MF-LAL, are multi-fidelity. Pocket2Mol and DecompDiff do not utilize any oracle during generation. Evaluation of generated compounds from all baselines is done with ABFE.

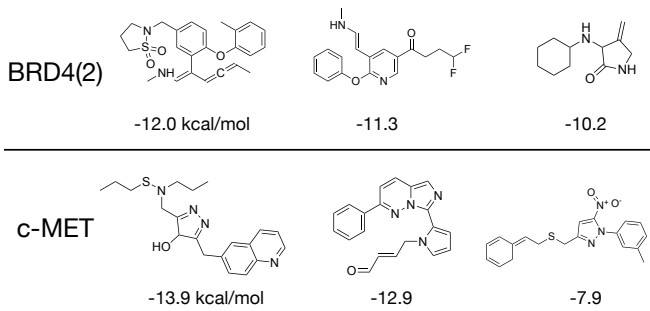

*Figure 3.* **Visualization of MF-LAL generated molecules.** The top 3 molecules and associated ABFE scores are shown for both the BRD4(2) (top row) and c-MET (bottom row) targets. The compounds are diverse and relatively drug-like.

### 4.3. Results

Table 1 reports the average and top 3 ABFE scores of 15 compounds generated by MF-LAL, as well as those generated by the baseline methods, following active learning for 7 days. We ran each method separately for two targets, BRD4(2) and c-MET. For MF-LAL and the most competitive baseline for each target (evaluated by lowest mean ABFE score), we evaluated an additional 25 compounds (total 40) for improved robustness, which is shown in Table 2. We also report the number of active scaffolds, which means the number of generated compounds that are active and have less than 0.4 Tanimoto similarity to any other active compound. We defined active as $< -8.2$ kcal/mol ($< 1\mu M$ activity) for BRD4(2) and $< -6.8$ kcal/mol ($< 10\mu M$ activity) for c-MET based on the affinities of compounds from previous works that investigate these targets (Liu et al., 2007; 2017; Naguib et al., 2024). Specifically, Liu et al. (2017) consider BRD4(2) inhibitors active when they have submicromolar activity, and Naguib et al. (2024) state that "potent activity" against c-MET is achieved with a 12 $\mu M$ inhibitor. The number of active scaffolds metric closely reflects the goal of early-stage drug discovery, which is to identify a diverse set of highly active compounds.

We filtered generated compounds such that all had QED (Bickerton et al., 2012) $> 0.4$, SAscore (Ertl & Schuffenhauer, 2009) $< 4$, and no rings with $\geq 7$ atoms. 55% of all compounds generated for BRD4(2), and 68% for c-MET, fulfilled these criteria. We also only allowed compounds that fit these criteria to be queried during active learning. We filtered compounds that did not meet the criteria after generation, instead of performing multi-objective optimization, because most generated compounds from single-objective optimization already had a QED/SAscore in the range of typical drug compounds (Bickerton et al., 2012; Ertl & Schuffenhauer, 2009). Following this filtering step, the mean QED of generated compounds for BRD4(2) (c-MET) was 0.59 (0.63), SAscore was 3.6 (3.5), and diversity (1 - mean pairwise Tanimoto similarity) was 0.81 (0.83).

*Table 1.* **Evaluation of 15 generated compounds at highest fidelity.** The mean and top 3 ABFE values are shown for 15 compounds sampled from each method after active learning for 7 days. "# active scafs." is the number of compounds ("scaffolds") that are active ($< -8.2$ kcal/mol for BRD4(2) and $< -6.8$ for c-MET) and have $< 0.4$ Tanimoto similarity to any other active compound.

| METHOD | BRD4(2) ABFE (KCAL/MOL) | | | | | | c-MET ABFE (KCAL/MOL) | | | | | |
|---|---|---|---|---|---|---|---|---|---|---|---|---|
| | | # ACTIVE | | | | | | # ACTIVE | | | | |
| | MEAN ± SD | SCAFS. | COUNT | 1ST | 2ND | 3RD | MEAN ± SD | SCAFS. | COUNT | 1ST | 2ND | 3RD |
| SF-VAE (ONLY ABFE) | -0.9 ± 2.7 | 0 | 15 | -5.7 | -2.9 | -2.9 | -1.2 ± 3.0 | 0 | 15 | -4.4 | -3.9 | -3.1 |
| SF-VAE (ONLY DOCKING) | -3.1 ± 2.8 | 0 | 15 | -6.1 | -5.3 | -4.8 | -2.8 ± 3.4 | 0 | 15 | -5.9 | -5.8 | -5.1 |
| REINVENT (ONLY ABFE) | -3.9 ± 3.4 | 2 | 15 | -8.7 | -8.3 | -8.2 | -2.9 ± 3.7 | 0 | 15 | -6.5 | -5.8 | -5.1 |
| REINVENT (ONLY DOCKING) | -3.1 ± 4.9 | 1 | 15 | -11.0 | -6.2 | -5.7 | -2.6 ± 5.0 | 1 | 15 | -8.0 | -6.8 | -5.9 |
| RGA (ONLY DOCKING) | -3.1 ± 3.9 | 0 | 15 | -7.8 | -7.0 | -6.8 | -2.1 ± 3.0 | 0 | 15 | -6.0 | -5.5 | -5.4 |
| VAE + 4x SF-GP | -2.3 ± 3.1 | 0 | 15 | -8.0 | -5.5 | -5.3 | -1.8 ± 2.5 | 0 | 15 | -6.3 | -5.9 | -5.1 |
| VAE + MF-GP | -1.3 ± 3.3 | 0 | 15 | -4.9 | -3.1 | -2.0 | -3.3 ± 2.9 | 1 | 15 | -9.7 | -6.0 | -4.2 |
| MF-AL-GFN | -2.5 ± 2.2 | 0 | 15 | -6.5 | -5.8 | -5.1 | -3.1 ± 1.8 | 0 | 15 | -5.5 | -4.5 | -4.1 |
| MF-AL-PPO | -2.8 ± 2.5 | 1 | 15 | -9.2 | -6.5 | -5.2 | -4.2 ± 2.8 | 0 | 15 | -6.6 | -5.8 | -5.5 |
| MF-GP + ZINC250K | -3.0 ± 2.9 | 0 | 15 | -5.5 | -4.7 | -4.5 | -2.9 ± 3.1 | 0 | 15 | -6.3 | -5.8 | -5.0 |
| POCKET2MOL | -4.3 ± 3.8 | 1 | 15 | -9.8 | -8.7 | -8.0 | -2.2 ± 4.2 | 0 | 15 | -4.5 | -3.9 | -3.2 |
| DECOMPDIFF | -2.7 ± 4.0 | 1 | 15 | -8.9 | -8.1 | -7.5 | -1.9 ± 6.4 | 1 | 15 | -8.0 | -5.1 | -2.7 |
| TAGMOL (ONLY DOCKING) | -1.9 ± 3.0 | 0 | 15 | -8.1 | -7.6 | -6.9 | -3.5 ± 3.3 | 1 | 15 | -7.0 | -5.6 | -5.1 |
| MF-LAL (OURS) | **-6.2** ± 3.9 | **6** | 15 | **-12.0** | **-10.2** | **-9.8** | **-6.7** ± 3.1 | **4** | 15 | **-12.9** | **-7.9** | **-7.7** |

*Table 2.* **Evaluation of 40 generated compounds at highest fidelity.** For MF-LAL and the most competitve baseline (Table 1), we ran ABFE on an additional 25 generated compounds for increased robustness. * refers to $p < 0.05$.

| METHOD | BRD4(2) ABFE (KCAL/MOL) | | | | | | c-MET ABFE (KCAL/MOL) | | | | | |
|---|---|---|---|---|---|---|---|---|---|---|---|---|
| | | # ACTIVE | | | | | | # ACTIVE | | | | |
| | MEAN ± SD | SCAFS. | COUNT | 1ST | 2ND | 3RD | MEAN ± SD | SCAFS. | COUNT | 1ST | 2ND | 3RD |
| MF-AL-PPO | | | | | | | -4.3 ± 2.6 | 1 | 40 | -8.5 | -6.6 | -5.8 |
| POCKET2MOL | -4.6 ± 3.8 | 2 | 40 | -9.8 | -9.8 | -9.3 | | | | | | |
| MF-LAL (OURS) | **-6.3*** ± 3.7 | **8*** | 40 | **-12.0** | **-11.3** | **-10.2** | **-7.1*** ± 3.0 | **6*** | 40 | **-13.9** | **-12.9** | **-7.9** |

We find that the average ABFE scores of the compounds generated by MF-LAL, as well as those of the top three compounds, are significantly better (lower kcal/mol) than the corresponding scores of compounds generated by the baseline methods for both targets. The difference in average ABFE score (predicted binding free energy) between MF-LAL and the top baseline is -1.6 kcal/mol for BRD4(2) ($p = 0.03$) and -2.8 kcal/mol for c-MET ($p < 0.001$), which are significant margins (see Appendix B.3 for details on the statistical tests). Additionally, MF-LAL generates significantly more active scaffolds than the most competitive baseline ($p < 0.05$ for both targets), and the mean ABFE score of the top 3 compounds from MF-LAL is significantly lower than the mean ABFE score of the top 3 compounds from baseline methods ($p < 0.05$ for both targets). Thus, our method outperforms both single and multi-fidelity techniques, as well as 3D structure-based methods (Pocket2Mol (Peng et al., 2022), DecompDiff (Guan et al., 2023b), and TAGMol (Dorna et al., 2024)) that do not use any binding affinity oracle. This suggests MF-LAL is most capable at generating compounds with real-world activity, since ABFE scoring is the gold standard prediction method. Note, too,

that the multi-fidelity techniques other than MF-LAL performed mostly similarly to the single-fidelity methods. This suggests that successfully taking advantage of multiple fidelities requires an architecture which, like that used in MF-LAL, is tailored to generating compounds for multiple fidelity oracles.

The molecular structures of the top 3 compounds generated by MF-LAL for both targets are shown in Figure 3. The compounds are structurally diverse, and none of them have close similars in the training set or in large datasets like PubChem (Kim et al., 2023). This shows the ability of MF-LAL to generate promising and novel structures. There are some similar ring structures across the compounds generated for c-MET, but this is not necessarily undesirable as it shows the model has found a high-property region of chemical space. The scaffold may be a promising starting point for development of empirical structure-activity relationships and lead optimization by medicinal chemists. Additionally, the mean pairwise Tanimoto similarity among the 40 generated compounds is $< 0.2$ for both targets, further indicating that our method generates a structurally diverse set.

*Table 3.* **Ablations on model architecture.** The mean and top 3 ABFE-computed energies are shown for 15 compounds sampled from each method after active learning for 7 days.

| METHOD | BRD4(2) ABFE (KCAL/MOL) | | | | | C-MET ABFE (KCAL/MOL) | | | | |
| | MEAN ± SD | # ACTIVE SCAFS. | 1ST | 2ND | 3RD | MEAN ± SD | # ACTIVE SCAFS. | 1ST | 2ND | 3RD |
|---|---|---|---|---|---|---|---|---|---|---|
| MF-LAL (OURS) | **-6.2** ± 3.9 | **6** | **-12.0** | **-10.2** | **-9.8** | **-6.7** ± 3.1 | **4** | **-12.9** | **-7.9** | **-7.7** |
| -FID. 1 | -6.1 ± 0.7 | 0 | -7.7 | -7.6 | -7.4 | -6.0 ± 1.1 | 1 | -8.8 | -7.0 | -6.0 |
| -FID. 2 | -5.1 ± 2.0 | 1 | -8.5 | -6.5 | -6.0 | -5.2 ± 2.5 | 2 | -8.0 | -7.3 | -6.1 |
| -FID. 3 | -4.2 ± 3.1 | 1 | -9.2 | -5.9 | -5.7 | -4.2 ± 3.5 | 1 | -9.8 | -7.1 | -6.1 |
| -FID. 4 | -2.4 ± 3.2 | 1 | -8.6 | -4.3 | -3.4 | -3.1 ± 3.0 | 1 | -7.6 | -6.7 | -5.1 |
| W/O LIKELIHOOD TERM | -3.4 ± 4.1 | 2 | -11.9 | -9.7 | -9.0 | -3.8 ± 3.7 | 2 | -10.9 | -7.7 | -6.3 |
| TRANSFORMER ENC/DEC | -6.1 ± 3.8 | 3 | -11.5 | -9.9 | -9.0 | -6.5 ± 2.9 | 2 | -11.6 | -7.6 | -6.5 |
| GCN ENC/DEC | -5.9 ± 3.0 | 2 | -10.9 | -10.1 | -9.0 | -6.1 ± 4.6 | 1 | -11.1 | -7.5 | -6.5 |

Table 3 reports results from various ablations of the MF-LAL architecture (using 15 compounds sampled from all methods). We experimented with removing each fidelity level individually, removing the likelihood term from the generation objective, and replacing the fully-connected encoder/decoder networks with a Transformer (Vaswani et al., 2017) and graph convolutional network (GCN for encoder and inner product decoder as described by Kipf & Welling (2016)). The results show that all fidelities contribute to performance. Removing the likelihood term greatly reduced the performance of our method, showing that the approach of only querying compounds at higher fidelities that also scored well at lower fidelities is critical for strong performance. Replacing the fully-connected encoder/decoder with a Transformer had little effect on performance, so we used the simpler fully-connected version. Finally, changing the molecular representation to a graph and replacing the fully-connected encoder/decoder with GCNs (Kipf & Welling, 2016) resulted in slightly worse performance.

## 5. Discussion and Conclusion

We present Multi-Fidelity Latent space Active Learning (MF-LAL), an integrated framework for generative and multi-fidelity surrogate modeling. Our experiments show that MF-LAL generates compounds with significantly higher activity, as predicted by a gold-standard binding free energy oracle, than other single and multi-fidelity approaches. Thus, MF-LAL shows promise as a way to generate compounds with real-world binding while incurring a reasonable computational cost.

Limitations of our approach include a limited set of oracles and a potential lack of synthesizability of the generated compounds, since SAscore is known to be imperfect (Skoraczyński et al., 2023). The SVGP technique we use for our surrogate models is known to overestimate the posterior variance far away from the inducing points (Bauer et al., 2016), potentially biasing our method towards out-of-distribution

molecules and increasing the unpredictability of the surrogate outputs. We also note that by design, MF-LAL may miss scaffolds at the highest fidelity that have strong binding if the lower fidelity oracles did not score that scaffold well. We consider this a necessary tradeoff to make the search for good compounds at the highest fidelity computationally tractable. Finally, it should also be noted that our experimental setup of evaluating each method after a fixed run time of 1 week may lead to high variance in the results, because each method has not been trained to convergence. While this most accurately reflects real-world usage, it may mean the ABFE values of the top generated compounds are somewhat variable between runs. Future work could include using a reaction-aware generative model that generates more synthesizable molecules (Horwood & Noutahi, 2020).

## Impact Statement

Similar to other works that apply machine learning to drug discovery, our work is subject to dual use (Urbina et al., 2022). There is potential for societal benefit, by helping develop new drug compounds to treat disease. However, there is also potential for harm, such as to generate new chemical weapons. Fortunately, the latter places a whole additional set of requirements on compounds (e.g. skin-absorbable or volatile and subject to inhalation), so this problematic direction does not appear to be imminent.

## Acknowledgments

This work was supported in part by NSF Grants #2205093, #2146343, #2134274, CDC-RFA-FT-23-0069, the U.S. Army Research Office under Army-ECASE award W911NF-07-R-0003-03, the U.S. Department Of Energy, Office of Science, IARPA HAYSTAC Program, DARPA AIE FoundSci and DARPA YFA. MKG has an equity interest in and is a cofounder and scientific advisor of VeraChem LLC.

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

# A. Model details

The encoder, decoder, and $h$ networks are all 3-layer feed-forward networks with ReLU activations and a 512-dimensional hidden layer. Each latent space has 64 dimensions. Molecules are represented using SELFIES strings (Krenn et al., 2020), and fed to the network using a flattened one-hot encoding representation. The loss on the latent encodings is the ELBO, which is the sum of the KL divergence and the cross-entropy loss of the reconstruction.

The surrogate models consist of a 4-layer deep kernel (Wilson et al., 2016) to encode the input and a Matern kernel for the covariance function. To accelerate training, we use an approximate GP trained with the ELBO loss (Hensman et al., 2015).

At each active learning step, we train the whole model from scratch until convergence with the Adam optimizer using a learning rate of 0.0001. For the molecule generation procedure using gradient-based optimization, we use the Adam optimizer with a learning rate of 0.1 for 100 epochs.

## A.1. Training details

Given a training molecule $x$ and associated oracle output $y$ at fidelity $k$, we minimize

$$L(\phi, \xi_{k-1}, \theta_k, \lambda_k; k, x, y) = \mathbb{E}_{\mathbf{z}_k \sim g(\cdot|x)} \log \frac{p_{\theta_k}(x, \mathbf{z}_k)}{g(\mathbf{z}_k|x)} + \int p(y|\hat{f}_k(\mathbf{z}_k)) p(\hat{f}_k(\mathbf{z}_k)|\mathbf{z}_k) d\hat{f}_k,$$

$$\text{where } g(\mathbf{z}_k|x) = \begin{cases} q_\phi(\mathbf{z}_k|x) & \text{if } k = 1 \\ h_{\xi_{k-1}}(\mathbf{z}_{k-1}) & \text{else} \end{cases} \tag{2}$$

where the first term is equivalent to maximizing the ELBO and the second is the MLL of the GP. While the loss is only evaluated at fidelity $k$, it is backpropagated through to all lower fidelities.

For the likelihood term of the generation objective, we maximize the likelihood of a point $\mathbf{z}_k^{(i)}$ in latent space $k$. This is equivalent to maximizing the probability density function evaluated at that point:

$$P\left(\mathbf{z}_k^{(i)}|\{(\mu_k^{(j)}, \sigma_k^{(j)})\}_{j=1}^M\right) = \sum_{j=1}^M \frac{1}{\sqrt{2\pi(\sigma_k^{(j)})^2}} \exp\left(-\frac{(\mathbf{z}_k^{(i)} - \mu_k^{(j)})^2}{2(\sigma_k^{(j)})^2}\right) \tag{3}$$

Additionally, we add a diversity term to the generation objective. We aim to minimize the following function:

$$\frac{1}{M^2} \sum_{i=1}^M \sum_{j=1}^M s(\mathbf{z}_k^{(i)}, \mathbf{z}_k^{(j)}) \tag{4}$$

where $s(A, B)$ is the cosine similarity between vectors $A$ and $B$.

# B. Experimental details

All experiments were conducted on a server with 8 RTX 2080 Ti GPUs. For our model and each baseline, we performed a random hyperparameter search with 20 trials using only the first 3 fidelities, and took the set of hyperparameters with the best generated samples at $f_3$ following 3 hours of active learning. We excluded ABFE from the hyperparameter search due to computational cost, and just used the same set of hyperparameters for all subsequent experiments using all 4 fidelities. We note that MF-LAL was somewhat sensitive to choice of hyperparameters, especially the KLD/reconstruction ratio for the decoders and the diversity coefficient used during generation. Thus, we advise users of MF-LAL to conduct a hyperparameter search similar to the method we used before applying MF-LAL to a new set of oracles.

Each model was provided with an initial dataset of random ZINC250k (Irwin et al., 2012) compounds evaluated at each fidelity. Each fidelity had 5 random compounds selected, except for the first fidelity, where we supplied 200,000 compounds and associated oracle outputs. This was because we wanted a large dataset of compounds to train the encoder and decoder at the first fidelity level, ensuring that generated compounds were reasonable, and running $f_1$ was almost instantaneous. For the more expensive oracles, however, we let active learning generate compounds to query to most efficiently use computational resources.

## B.1. Oracles

For all oracles, we estimated the cost (for the baselines that require it) using the average run time over 10 samples with random input compounds. We also computed the ROC-AUC (shown below) of each oracle for the BRD4(2) and c-MET targets to confirm that the higher cost oracles are more accurate. To do this, we generated a dataset of 32 known active and presumed inactive compounds for BRD4(2) and c-MET, and then ran each oracle on all of the compounds. The actives were retrieved from the BRD4(2)/c-MET target from BindingDB with $K_i < 10\mu M$, and the inactives were generated using DUD-E (Mysinger et al., 2012). As expected, for both targets, the ROC-AUC increases with higher fidelity, as well as the computational cost. This shows that the higher cost oracles are indeed more accurate, and so the hierarchy of fidelities we use is sensible and applicable to multi-fidelity learning.

**Linear regression (cost: 0.1s, ROC-AUC BRD4(2): 0.59, c-MET: 0.68)**    We used BRD4(2) and, separately, c-MET data from BindingDB (Liu et al., 2007) to train a simple linear regression model. The input to the model was 2048-bit Morgan fingerprints, and the output was the experimental binding energy in $K_i$, converted to kcal/mol.

**AutoDock4 (cost: 4s, ROC-AUC BRD4(2): 0.73, c-MET: 0.72)**    We prepared the AutoDock4 grid files using AutoDock-Tools (Morris et al., 2009). Arbitrary ligands were prepared using obabel (O'Boyle et al., 2011) with pH 7.4 and gasteiger partial charges. We used AutoDock-GPU (Santos-Martins et al., 2021), a GPU-accelerated version of AutoDock4, for all computation. For each ligand, we performed 20 random restarts and selected the minimum predicted energy.

**Ensembled AutoDock4 (cost: 44s for BRD4(2) and 68s for c-MET, ROC-AUC BRD4(2): 0.80, c-MET: 0.80)**    Same as above, except we used the minimum energy from 8 separate AutoDock4 runs using the same ligand and each of the following protein crystal structures (listed as PDB IDs): `5ues`, `5uet`, `5uev`, `5uez`, `5uf0`, `5uvs`, `5uvy`, `5uvz` for BRD4(2), and `2wd1`, `4deg`, `4dei`, `4r1v`, `5eob` for c-MET

**Absolute binding free energy (ABFE) (cost: 9:20hrs, ROC-AUC BRD4(2): 0.92, c-MET: 0.89)**    We use the Binding Affinity Tool (BAT.py) implementation (Heinzelmann & Gilson, 2021) for absolute binding free energy calculation, available at `https://github.com/GHeinzelmann/BAT.py`. For BRD4(2), we use the short **tevb** calculations, which were introduced recently to reduce computational cost (Heinzelmann et al., 2024). For c-MET, we use the standard SDR method, but we found we could reduce simulation times for all components to 30% of their original amounts and still retain strong performance. All molecular dynamics simulators are run with AMBER with GPU support. As BAT.py requires a starting pose for the ligand, we used the pose generated from AutoDock4. We additionally wrote custom scripts to parallelize molecular dynamics runs across all available GPUs.

## B.2. Baselines

The details of each baseline are as follows:

- **SF-VAE (only ABFE / only docking)** (Gómez-Bombarelli et al., 2018). The VAE encoder and decoder, and GP surrogate, are set up similarly to those in MF-LAL. The upper confidence bound is used as an acquisition function.

- **REINVENT (only ABFE / only docking)** (Olivecrona et al., 2017). Used the code available at `https://github.com/MarcusOlivecrona/REINVENT`.

- **RGA (only docking)** (Fu et al., 2022). Used the code available at `https://github.com/futianfan/reinforced-genetic-algorithm`.

- **VAE + 4x SF-GP**. The VAE encoder and decoder, and GP surrogates, are set up similarly to those in MF-LAL. We also use the same acquisition function and generative procedure as MF-LAL for this baseline, except without the need to map points between latent spaces.

- **VAE + MF-GP**. We use the Multi-Fidelity Max Value Entropy acquisition function for selecting compounds during active learning (Takeno et al., 2020), and a linear truncated fidelity kernel (Gardner et al., 2018) for the GP surrogate.

- **MF-AL-GFN** (Hernandez-Garcia et al., 2023) Used the code available at `https://github.com/nikita-0209/mf-al-gfn`.

- **MF-AL-PPO** (Hernandez-Garcia et al., 2023). Provided in the same codebase as the one referenced above.

- **MF-GP + ZINC250k**. We use the Multi-Fidelity Max Value Entropy acquisition function for selecting compounds during active learning (Takeno et al., 2020), and a linear truncated fidelity kernel (Gardner et al., 2018) for the GP surrogate. All ZINC250k (Irwin et al., 2012) compounds are provided as the candidate set.

- **Pocket2Mol** (Peng et al., 2022). Used the code available at `https://github.com/pengxingang/Pocket2Mol`.

- **DecompDiff** (Guan et al., 2023b). Used the code available at `https://github.com/bytedance/DecompDiff`.

- **TAGMol (only docking)** (Dorna et al., 2024). Used the code available at `https://github.com/MoleculeAI/TAGMol`. We used the single-objective gradient guidance model for optimizing binding affinity.

### B.3. Statistical tests

To compare the mean ABFE score of generated compounds from each method, we used a standard Student's t-test. We also used a standard t-test for the mean of the top 3 generated compounds from each method. For these tests, we used the entire set of 40 compounds if we had run them for a given method, otherwise we used 15. We only analyzed the differences in the top 3 compounds among the methods that had 40 compounds. For comparing the number of active scaffolds, we used a binomial test to compare the proportion of distinct active compounds among the total generated compounds. For this test, we cannot directly compare the samples of 40 compounds with the samples of 15 compounds, because we expect the number of active scaffolds to plateau as we generate more compounds (meaning a proportion test across samples with different sizes is not valid). Therefore, the p-value we report for number of active scaffolds only compares MF-LAL with the top baseline. However, we found that limiting all methods to 15 generated compounds still showed statistically significant results favoring MF-LAL.

## C. Additional results

### C.1. Oracle outputs during active learning

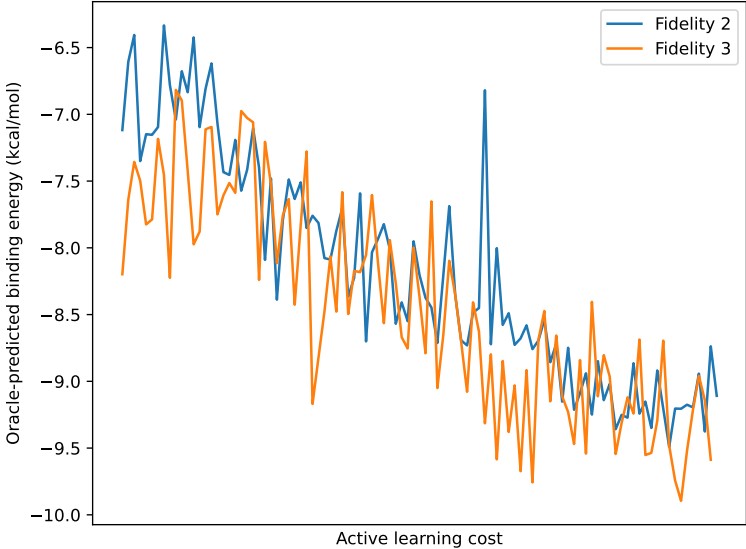

*Figure 4.* **Oracle outputs during active learning**. The y-axis shows the oracle-predicted binding energy of the generated query compounds, and the x-axis shows active learning progress.

Figure C.1 shows the oracle-predicted binding energy of the compounds generated during the MF-LAL active learning process. We only show fidelities 2 and 3, because fidelity 1 is already supplied with a large initial dataset so there is little improvement in the query quality during active learning, and fidelity 4 does not show any noticeable improvement due to a relatively small number of queries made. For fidelities 2 and 3, we observe a marked improvement of the predicted binding energy over active learning, showing that MF-LAL successfully learns what makes a compound favorable at each fidelity level.

## C.2. Reconstruction accuracy

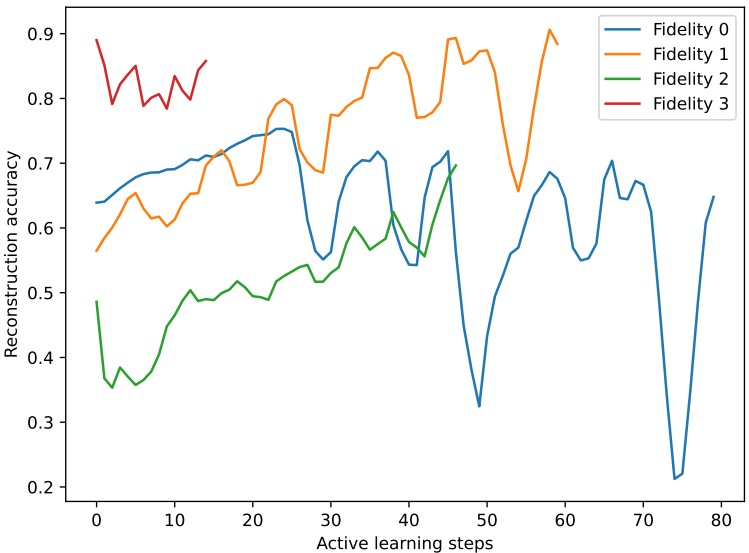

*Figure 5.* **Reconstruction accuracy during active learning**. The y-axis shows the proportion of training set compounds that were successfully reconstructed using the decoder, and the x-axis shows the active learning iteration.

Figure C.2 shows the reconstruction accuracy of the decoders during active learning. Reconstruction accuracy is the proportion of compounds that were exactly reconstructed, meaning every SELFIES character is identical to the input. The decoder corresponding to the highest fidelity latent space achieves the overall highest reconstruction accuracy, which is likely because it only has to decode from a very limited compound space. The lowest fidelity decoder has the worst reconstruction accuracy, because it decodes the most varied set of compounds. Nonetheless, the reconstruction accuracies are relatively high across all decoders, meaning MF-LAL successfully learns the mapping from latent space back to molecule at all fidelities.

