# OpenReview forum: "MF-LAL: Drug Compound Generation Using Multi-Fidelity Latent Space Active Learning"
_ICML.cc/2025/Conference — ICML 2025 poster_

### Official Review · Reviewer_QauA · 2025-03-07

**Overall Recommendation:** 3

**Summary:**

The paper introduces a novel framework for multi-fidelity active learning that is based on learning of hierarchical latent representations. The authors evaluate the approach in a multi-fidelity setting culminating in ABFE evaluation on two different protein targets.

**Claims And Evidence:**

The claims are supported by evidence, with some questions regarding the experimental evaluation and presentation of the results (below).

**Essential References Not Discussed:**

N/A

**Experimental Designs Or Analyses:**

Discussed below.

**Methods And Evaluation Criteria:**

Used multi-fidelity setting, in particular the choice of oracles, is convincing, and the method is evaluated in two different protein targets, which further strengthens the results. The main issue lies in the number of tested molecules (discussed in more detail below).

**Other Comments Or Suggestions:**

- “We will focus on “query synthesis” approaches (Angluin, 1988), where the model generates its own queries to send to the oracles, speeding up learning compared to approaches that query oracles with samples from a fixed candidate set.” - I’d be curious to see some citation backing up this claim (that’s also more recent).
- “Ensembled AutoDock4 (cost: 44s, ROC-AUC BRD4(2): 0.80, c-MET: 0.80)” - one cost is provided for both targets, while c-MET seems to use 5 crystal structures, whereas the other uses 8; could authors explicitly confirm that the cost is supposed to be the same in both cases?
- “Additionally, the pairwise Tanimoto similarity among the 40 compounds generated by MF-LAL is less than 0.2” - does this refer to mean pairwise similarity?

**Other Strengths And Weaknesses:**

The problem setting is of (in my opinion) high significance, and the authors thoroughly evaluate their proposed method, including some ablations and comparison with reasonable baselines. The setup of oracles also reflects real-life setting, which is another strength of the paper.

The biggest weakness, in my opinion, is however also related to this. Because the authors opt to use a setting close to real-life, with MD as the highest-fidelity oracle, they are limited to sampling very few samples in the end (15 per method). This is somehow understandable given high computational cost, but limits the reliability of the results. I wonder if it would be possible to instead do a simulated multi-fidelity setup, as in some earlier papers; or perhaps to pick a lower-cost highest-fidelity oracle. I do not expect the authors to necessarily address this in any way, since it would require a substantial rework of the experiments, but raising my concern about the reliability of the presented results.

About actionable weaknesses, I do have the following comment:

“For MF-LAL and the most competitive baseline for each target, we ran an additional 25 compounds” - this seems to me to be a bizarre design decision, since the authors compare, among other things, the number of discovered modes (“scaffolds”). This obfuscates the results, in particular since for the scaffolds, the authors perform statistical analysis only for two best methods (the information about which is hidden away in the appendix), which might be misleading. I would strongly suggest limiting the results in this table to the same budget of oracle calls (15 each), and perhaps having a separate table / figure for 40 compound setting.

**Questions For Authors:**

N/A

**Relation To Broader Scientific Literature:**

The paper builds on earlier work on multi-fidelity generative models, including MF-GFN, and expends on some of its perceived limitations.

**Theoretical Claims:**

N/A

---

> ### Author Rebuttal · Authors · 2025-03-31
>
> We thank the reviewer for their helpful feedback and positive comments about the work.
>
> > **Q1:** Because the authors opt to use a setting close to real-life, with MD as the highest-fidelity oracle, they are limited to sampling very few samples in the end (15 per method). This is somehow understandable given high computational cost, but limits the reliability of the results.
>
> **A1:** We would like to emphasize the statistical tests in Section 4.3 and Appendix B.3, where we show that the ABFE scores of MF-LAL compounds are significantly better than those from baselines in terms of both mean and top scores for both targets. So while we do have a relatively small number of samples, we show that MF-LAL makes statistically significant and reliable improvements over baselines.
>
> > **Q2:** I wonder if it would be possible to instead do a simulated multi-fidelity setup, as in some earlier papers; or perhaps to pick a lower-cost highest-fidelity oracle.
>
> **A2:** While many previous works have indeed focused on a simulated multi-fidelity setup, we do not think that such a setup is a very good proxy for real-world simulators. This is because the method of ABFE calculation is fundamentally different from docking or other activity prediction techniques, and so it is difficult to substitute it with other cheaper techniques and still reach useful conclusions. If it is possible to reach statistically significant conclusions using the real-world MD simulator, which we believe to have done in this paper, then we think it is most valuable to use the real simulator instead of an artificial setup.
>
> > **Q3:** ...for the scaffolds, the authors perform statistical analysis only for two best methods (the information about which is hidden away in the appendix), which might be misleading. I would strongly suggest limiting the results in this table to the same budget of oracle calls (15 each), and perhaps having a separate table / figure for 40 compound setting.
>
> **A3:** We agree that making two separate tables for 15 and 40 compounds would increase the clarity of the results. In the updated draft, we will include these two tables, as well as statistical tests comparing MF-LAL to all applicable baselines in both settings. As noted in Appendix B.3, statistical tests comparing MF-LAL to all baselines where all methods were limited to 15 compounds still showed statistically significant results in favor of MF-LAL (except for the binomial test for the number of active scaffolds between MF-LAL, which generated 4 scaffolds, and REINVENT, MF-AL-PPO, and DecompDiff which generated 1 scaffold, where p=0.07). The results for MF-LAL and the top baseline for each target using only 15 compounds are shown below, because they were not included in the original draft. These results remain consistent with our conclusions about the strong performance of MF-LAL.
>
> | | BRD4(2) ABFE | | | | | | c-MET ABFE | | | | | |
> | --- | --- | ---| --- | --- | --- | --- | --- | --- | --- | --- | --- | --- |
> | Method | Mean $\pm$ std | # active scafs | Count | 1st | 2nd | 3rd | Mean $\pm$ std | # active scafs | Count | 1st | 2nd | 3rd |
> | MF-AL-PPO | | | | | | | -4.2 $\pm$ 2.8 | 0 | 15 | -6.6 | -5.8 | -5.5 |
> | Pocket2Mol | -4.3 $\pm$ 3.8 | 1 | 15 | -9.8 | -8.7 | -8.0 | | | | | | |
> | MF-LAL | **-6.2** $\pm$ 3.9 | **6** | 15 | **-12.0** | **-10.2** | **-9.8** | **-6.7** $\pm$ 3.1 | **4** | 15 | **-12.9** | **-7.9** | **-7.7** |
>
>
>
> > **Q4:** I’d be curious to see some citation backing up this claim [about query synthesis] (that’s also more recent).
>
> **A4:** [1, 2, 3, 4] are recent papers that explore query synthesis approaches, and many of them find that query synthesis outperforms traditional pool-based methods in active learning. We will include these citations in the updated draft.
>
> [1] Morand et al. “Efficient Exploration of Microstructure-Property Spaces via Active Learning.” Frontiers in Materials 2022.
>
> [2] Schumann and Rehbein. “Active Learning via Membership Query Synthesis for Semi-Supervised Sentence Classification.” CoNLL 2019.
>
> [3] Guo et al. “Dual generative adversarial active learning.” Applied Intelligence 2021.
>
> [4] Zhu and Bento. “Generative Adversarial Active Learning.” arXiv 2017.
>
> > **Q5:** “Ensembled AutoDock4 (cost: 44s, ROC-AUC BRD4(2): 0.80, c-MET: 0.80)” - one cost is provided for both targets, while c-MET seems to use 5 crystal structures, whereas the other uses 8; could authors explicitly confirm that the cost is supposed to be the same in both cases?
>
> **A5:** Thank you for pointing out this omission, the shown cost is indeed only for BRD4(2). The correct cost for c-MET is 68s, which we will include in the updated draft.
>
> > **Q6:** “Additionally, the pairwise Tanimoto similarity among the 40 compounds generated by MF-LAL is less than 0.2” - does this refer to mean pairwise similarity?
>
> **A6:** Yes, mean pairwise similarity. We will update the draft to clarify.

---

### Official Review · Reviewer_kXA7 · 2025-03-08

**Overall Recommendation:** 3

**Summary:**

This paper introduces Multi-Fidelity Latent space Active Learning(MF-LAL), an framework that  integrates a set of different oracle functions to guide the generation of molecules to get higher predicted activity. It combines the generative model and surrogate model into a single framework, and the computational cost is reduced with active learning method. MF-LAL is able to achieve around 50% improvement in binding free energy score for the molecules generated compared to baseline methods, especially for two disease-relevant proteins (BRD4(2) and c-MET).

**Claims And Evidence:**

The main claim of this paper is the effectiveness of the proposed MF-LAL framework, which surpass other baseline methods in molecule binding free energy optimization task. Here are some concerns:

1. In table 1, POCKET2MOL and MF-LAL generated 40 molecules while other baseline methods generated 15 molecules. It seems that this is unfair as generating more molecules would definitely result in better ABFE scores for the top-3 molecules and more active scaffolds. But the increase in mean value is solid, so I think the author should show the separated results mean value for 40 molecules ; number of active scaffolds and top-3 values for 15 molecules.

2. Another concern is that authors use Pocket2Mol and DecompDiff as baseline model. But they are 3D pocket-based molecule generation models instead of optimization models. There are some 3D optimization models, like, DecompOpt[1] and TagMol[2]. Also RGA[3] can be included


3. The case shown in Figure 2 are not very good. There are some uncommon or weird structures in the generated molecules.





[1] Zhou, Xiangxin, et al. "DecompOpt: Controllable and Decomposed Diffusion Models for Structure-based Molecular Optimization." The Twelfth International Conference on Learning Representations.

[2] Dorna, Vineeth, et al. "TAGMol: Target-Aware Gradient-guided Molecule Generation." ICML'24 Workshop ML for Life and Material Science: From Theory to Industry Applications.

[3] Fu, Tianfan, et al. "Reinforced genetic algorithm for structure-based drug design." Advances in Neural Information Processing Systems 35 (2022): 12325-12338.

**Essential References Not Discussed:**

As previously stated, 3D optimization models, like RGA[1], DecompOpt[2] and TagMol[3] should be referenced and add to baselines.


[1] Fu, Tianfan, et al. "Reinforced genetic algorithm for structure-based drug design." Advances in Neural Information Processing Systems 35 (2022): 12325-12338.

[2] Zhou, Xiangxin, et al. "DecompOpt: Controllable and Decomposed Diffusion Models for Structure-based Molecular Optimization." The Twelfth International Conference on Learning Representations.

[3] Dorna, Vineeth, et al. "TAGMol: Target-Aware Gradient-guided Molecule Generation." ICML'24 Workshop ML for Life and Material Science: From Theory to Industry Applications.

**Experimental Designs Or Analyses:**

The experimental design is reasonable. It is done on two cancer-relevant proteins, BRD4(2) and c-MET. Also, ABFE  are well-validated on those targets and have good agreement with experimental data.

As previously discussed, the issues are the limited number of targets, and the unmatched number

**Methods And Evaluation Criteria:**

## Method

The method proposed is interesting. It is tring to integrate multi-fidelity surrogate functions to guide the molecule generation. It introduces a Hierarchical Latent Space Representation to optimize the latent space at each fidelity and decode the molecule at highest fidelity. It also includes an active learning step. Overall I think the method is simple but effective and makes sense for the problem.

## Evaluation

The evaluation is done on two targets. The concern is that the number of targets is too limited, but it is understandable as ABEF is very time-consuming and expensive.

**Other Comments Or Suggestions:**

1. If possible, it would be benificial to have figure 3 shown in the main text as it gives a more clear illustration of the framework.

**Other Strengths And Weaknesses:**

## Other Strengths

1. The paper is well written, with clear description of methods and experiment results.

2. The method is simple but effective


## Other Weaknesses

1.  some 3d based optimization models are not included in the baseline

2.  tested targets are limited

3. when comparing the top-3 values, it should also generating 15 molecules as for other methods for fair comparasion

**Questions For Authors:**

1. From the ablation it seems that even the Linear regression fidelity contributes to the performance. Removing it would results in significant decrease of number of active scaffolds from 4 to 0. The ABFE scores for top 3 molecule also significantly decreased. Can you explain more on why this happens? My understanding is that it should have some kind of influence but should not be that obvious, as ABFE is much accurate than a linear regression model.

**Relation To Broader Scientific Literature:**

I think the scope of this paper is molecule optimization for better binding energy. It does not relate to the broader scientific literature.

**Theoretical Claims:**

There is no theoretical claims in this paper.

---

> ### Author Rebuttal · Authors · 2025-03-31
>
> We thank the reviewer for their helpful feedback and positive comments about the work.
>
> > **Q1:** In table 1, POCKET2MOL and MF-LAL generated 40 molecules while other baseline methods generated 15 molecules. It seems that this is unfair as generating more molecules would definitely result in better ABFE scores for the top-3 molecules and more active scaffolds. But the increase in mean value is solid, so I think the author should show the separated results mean value for 40 molecules ; number of active scaffolds and top-3 values for 15 molecules
>
> > when comparing the top-3 values, it should also generating 15 molecules as for other methods for fair comparasion
>
> **A1:** Based on this and other comments from reviewers, we will make two separate tables in the updated draft reporting results on both 15 (for all methods) and 40 compounds (for the methods that have them). See the response to Reviewer QauA for the MF-LAL and baseline results on only 15 compounds, including the top compound scores and the number of active scaffolds, which remain consistent with our conclusions about the strong performance of MF-LAL.
>
> > **Q2:** Another concern is that authors use Pocket2Mol and DecompDiff as baseline model. But they are 3D pocket-based molecule generation models instead of optimization models. There are some 3D optimization models, like, DecompOpt[1] and TagMol[2]. Also RGA[3] can be included
>
> **A2:** Thank you for the references, we will cite them in the updated draft. We agree that comparing MF-LAL to a 3D optimization model would be valuable, so we are currently running TAGMol as a baseline. While the results are not yet finished due to computational cost, we will post them when they are finished during the author-reviewer discussion period. Following the reviewer’s suggestion, we have also run RGA. We ran RGA using a single fidelity (docking) as the oracle, similar to how we implemented the SF-VAE (only docking) and REINVENT (only docking) baselines in our paper. We chose to use docking, instead of ABFE, as the single fidelity oracle due to computational cost and the presumed inability of a genetic algorithm to make use of a very small number of ABFE oracle calls. We used the default parameters of RGA, with random ZINC250k compounds as the starting population for the genetic algorithm. The results from RGA are as follows:
>
> | | BRD4(2) ABFE | | | | | | c-MET ABFE | | | | | |
> | --- | --- | ---| --- | --- | --- | --- | --- | --- | --- | --- | --- | --- |
> | Method | Mean $\pm$ std | # active scafs | Count | 1st | 2nd | 3rd | Mean $\pm$ std | # active scafs | Count | 1st | 2nd | 3rd |
> | RGA (only docking) | -3.1 $\pm$ 3.9 | 0 | 15 | -7.8 | -7.0 | -6.8 | -2.1 $\pm$ 3.0 | 0 | 15 | -6.0 | -5.5 | -5.4 |
> | MF-LAL | **-6.3** $\pm$ 3.7 | **8** | 40 | **-12.0** | **-11.3** | **-10.2** | **-7.1** $\pm$ 3.0 | **6** | 40 | **-13.9** | **-12.9** | **-7.9** |
>
> The results from MF-LAL are also shown for comparison. MF-LAL significantly outperforms RGA, with the latter having performance similar to that of the REINVENT baseline. We will include these results in the updated draft.
>
> Finally, we do not think DecompOpt is a good baseline to compare with because it requires knowledge of existing binders for generation (to compute the “reference arms”). Since MF-LAL and our other baselines are de novo generation methods that do not use knowledge from existing binders, we do not think a comparison with DecompOpt would be fair.
>
> > **Q3:** tested targets are limited
>
> **A3:** The number of targets we tested is limited by the high cost of ABFE calculations. In addition, the ABFE framework we use is only validated and configured for a few targets, and BRD4(2) and c-MET are the only of those targets that are of interest from a biological/drug discovery perspective.
>
> > **Q4:** From the ablation it seems that even the Linear regression fidelity contributes to the performance. Removing it would results in significant decrease of number of active scaffolds from 4 to 0. The ABFE scores for top 3 molecule also significantly decreased. Can you explain more on why this happens? My understanding is that it should have some kind of influence but should not be that obvious, as ABFE is much accurate than a linear regression model.
>
> **A4:** The lowest fidelity oracle is critical for performance because it provides the majority of the data for the multi-fidelity model. As stated in Appendix B, we provide the model with an initial dataset of 200,000 ZINC250k compounds and associated oracle outputs at the first fidelity level (linear regression). Thus, removing this fidelity greatly reduces the total data available to the model. Even if this data is significantly less accurate than ABFE, the quantity of data is still important for model performance, and so it is expected that removing this data significantly reduces performance.

---

### Official Review · Reviewer_rMi8 · 2025-03-13

**Overall Recommendation:** 3

**Summary:**

This paper introduces a new approach for generating drug candidates, called MF-LAL. The proposed method utilizes a variational autoencoder with multiple latent spaces arranged hierarchically to accommodate different fidelities. The first level employs a regression model trained to predict activity on known compounds. The second and third levels are based on molecular docking to one or several protein structures, respectively. The final level focuses on absolute binding free energy (ABFE) prediction, which is the most computationally intensive model for estimating binding to the target protein. Compounds are generated using an active learning method with query synthesis. The acquisition function relies on surrogate models trained on data already collected from various fidelities. Initially, molecules of the lowest fidelity are generated until uncertainty falls below a predefined threshold. Subsequently, the model begins generating molecules from the next level of fidelity. This process continues for seven days. The results show that MF-LAL can generate molecules with the best ABFE, outperforming single-fidelity and single-latent-space models.

## update after rebuttal

The Authors addressed all my comments. I decided to maintain my positive score.

**Claims And Evidence:**

The claims in the paper are supported by experimental results.

**Essential References Not Discussed:**

The key references have been described in the paper.

**Experimental Designs Or Analyses:**

The experimental design is sound, but a few aspects could be improved. For example, only 15 compounds are generated and evaluated in Tables 1 and 2 for all but two best models. I believe all methods should be assessed based on 40 compounds. It is unclear whether the number of active scaffolds and the results of the top compounds are computed from all 40 compounds or just 15. Furthermore, the initially generated compounds prior to filtering could be assessed in terms of validity, synthetic accessibility, and drug-likeness.

**Methods And Evaluation Criteria:**

The method is explained clearly, and the selected evaluation metrics for the two biological targets effectively demonstrate the value of the proposed model. However, the choice of the decoder architecture is nonstandard and is compared only with non-autoregressive methods (more details in Questions For Authors). There are no details on how GCN was implemented for generating molecules. Moreover, I am curious about the validity of the generated compounds. The SELFIES representation is used to ensure that the string representation can be decoded to molecules, but rather heavy filtering criteria are applied (QED > 0.4, SA < 4, no rings with at least seven atoms) - what percentage of the generated compounds match these criteria for the tested models?

**Other Comments Or Suggestions:**

In Algorithm 1, there is probably a typo in line 7. It should be $\Sigma_{\lambda_k}(z_k)<\gamma_k$.

**Other Strengths And Weaknesses:**

All of my comments are described in the other sections.

**Questions For Authors:**

1. How were thresholds -8.2 and -6.8 kcal/mol chosen for the two tested targets? Was it based on reference compounds or balance of the activity classes in training regression model for the lowest fidelity?
2. The choice of the decoder architecture seems nonstandard. Text representations such as SMILES and SELFIES are usually generated in the autoregressive fashion. Have you tried using RNNs? Was the transformer trained also to predict all characters at the same time, or autoregressively?

**Relation To Broader Scientific Literature:**

This paper presents an intriguing proposition for effectively training generative models to produce increasingly useful molecule candidates. A hierarchical approach with multiple latent spaces is proposed to capture representations for each fidelity separately. Stochastic variational Gaussian processes are used as surrogate functions trained on the obtained calculations of binding affinity, making these functions easy to train even for big datasets. MF-LAL can be useful in early drug discovery stages to propose novel hit candidates.

**Theoretical Claims:**

There are no proofs of theoretical claims that need to be checked.

---

> ### Author Rebuttal · Authors · 2025-03-31
>
> We thank the reviewer for their helpful feedback and positive comments about the work.
>
> > **Q1:** There are no details on how GCN was implemented for generating molecules.
>
> **A1:** We will include details of our GCN implementation in the Appendix in the updated draft. Briefly, we used a three-layer graph convolutional network with one-hot encoded atom types for the encoder, and an inner product decoder as described in Kipf and Welling 2016.
>
> > **Q2:** The SELFIES representation is used to ensure that the string representation can be decoded to molecules, but rather heavy filtering criteria are applied (QED > 0.4, SA < 4, no rings with at least seven atoms) - what percentage of the generated compounds match these criteria for the tested models?
>
> **A2:** Among all compounds generated after training with active learning, 55% for BRD4(2) and 68% for c-MET fulfilled the filtering criteria. We find these percentages high enough such that there is no need to do multi-objective optimization, especially because generation is very fast so discarding 25-50% of the generated molecules is not problematic. We will include these numbers in the updated draft.
>
> > **Q3:** It is unclear whether the number of active scaffolds and the results of the top compounds are computed from all 40 compounds or just 15.
>
> **A3:** The number of active scaffolds and the top 3 compounds in Table 1 are computed from all 40 compounds (for MF-LAL and the top baselines). Based on this and other comments from reviewers, we will make two separate tables in the updated draft reporting results on both 15 and 40 compounds for clarity. See the response to Reviewer QauA for the MF-LAL results on only 15 compounds, including the top compound scores and the number of active scaffolds.
>
> > **Q4:** the initially generated compounds prior to filtering could be assessed in terms of validity, synthetic accessibility, and drug-likeness.
>
> **A4:** Prior to filtering, the mean SAscore of generated compounds is 3.9 for BRD4(2) and 3.6 for c-MET. The mean QED is for 0.48 BRD4(2) and 0.50 for c-MET. The validity is 100% because we used SELFIES strings, which are guaranteed to be valid. We will include these numbers in the updated draft.
>
> > **Q5:** How were thresholds -8.2 and -6.8 kcal/mol chosen for the two tested targets? Was it based on reference compounds or balance of the activity classes in training regression model for the lowest fidelity?
>
> **A5:** The thresholds were chosen based on reference compounds from previous works that investigate BRD4(2) and c-MET (see lines 368-373, left side). We roughly picked these thresholds based on the typical experimental affinities of the best binders analyzed in these works. For BRD4(2), Liu et al. 2017 (see paper for citation) explore various BRD4(2) inhibitors and generally consider compounds active when they have submicromolar (<1 $\mu$M) activity, which is the cutoff we used in our paper. For c-MET, Naguib et al. 2024 states that “potent activity” is achieved with a 12 $\mu$M inhibitor, so we set our activity cutoff to <10 $\mu$M. We will include these details in the updated draft.
>
> > **Q6:** The choice of the decoder architecture seems nonstandard. Text representations such as SMILES and SELFIES are usually generated in the autoregressive fashion. Have you tried using RNNs? Was the transformer trained also to predict all characters at the same time, or autoregressively?
>
> **A6:** Both Transformers and RNNs have indeed been used in such applications. We chose to only test the Transformer architecture, however, because it has demonstrated very similar or slightly superior performance relative to RNNs on molecular generation tasks [1, 2]. Our Transformer was trained to predict characters autoregressively using the standard Transformer decoder architecture.
>
> [1] Chen et al.”Molecular language models: RNNs or transformer?” Briefings in Functional Genomics 2023.
>
> [2] Xu et al. “REINVENT-Transformer: Molecular De Novo Design through Transformer-based Reinforcement Learning.” arXiv 2024.

---

### Official Review · Reviewer_URfo · 2025-03-16

**Overall Recommendation:** 3

**Summary:**

This paper introduces, MF-LAL,  a generative algorithm for drug discovery based on biological activity rather than docking. Rather than conditioning on molecular docking, the authors propose a pipeline to generate molecules based on molecular dynamics-based binding free energy. As MD-based free energy calculations are prohibitively expensive, MF-LAL uses multiple oracles at varying fidelity levels to z. The authors also present a sample of efficient training to minimize the high-fidelity data required. The active learning-based method generates molecules based on the acquisition function over the hierarchical latent space and expands the dataset. Optimized molecules are then generated using gradient-based optimizations to find extrema in the latent space at some fidelity.

## Update After Rebuttal

I have read the rebuttal and decided to keep my score.

**Claims And Evidence:**

* A key component of the active learning method present is the threshold calculation in algorithm 2 line 7 $\sigma_{\lambda_k}(\cdot)$ must be well calibrated in order for the active learning to work.
* The reconstruction accuracy of the decoder is quite small and not very robust

**Essential References Not Discussed:**

N/A

**Experimental Designs Or Analyses:**

- The usual generation quality measures such as validity, diversity, and synthesizability are not presented
- Since all the baselines don’t have statistically significant results, it is a bit concerning to compare the results

**Methods And Evaluation Criteria:**

* The training details including the supplement are sparse and disjointed. It is not easy to understand how exactly the model is trained. I know space is limited, but even an algorithm in the supplement in addition to the loss function would help, especially for
* The hierarchical latent space is interesting and optimizing at a single fidelity updates the latent space in all fidelities
* One concern would be that such an optimization causes adverse effects on the fidelity not being trained.

**Other Comments Or Suggestions:**

* Line 114 Col 2: querying oracle k –> querying oracle f_k
* Line 209 Col 2: We use the posterior variance of the GP surrogate $\sigma_{\lambda_k}(\cdot)$ –> We use the posterior variance, $\sigma_{\lambda_k}(\cdot)$, of the FP surrogate $\hat{f}_k$

**Other Strengths And Weaknesses:**

* The retrieval component of the pipeline is very similar to fuzzy search algorithms and semantic search algorithms and is not particularly novel.
* Substructure similarity is measured with Tanimoto distance of Morgan fingerprints which does not capture 3D information.
* The authors have many baselines and discuss potential drawbacks due to the lack of experimental samples

**Questions For Authors:**

N/A

**Relation To Broader Scientific Literature:**

The authors tackle a very important problem of sample efficient training of surrogate modeling. It is usually prohibitively costly to acquire the amount of data usually needed for deep learning models. So a method that maximizes sample efficiency is hugely important in the specific field of drug discovery but also beyond to other scientific domains.

**Theoretical Claims:**

N/A

---

> ### Author Rebuttal · Authors · 2025-03-31
>
> We thank the reviewer for their helpful feedback and positive comments about the work.
>
> > **Q1:** The training details including the supplement are sparse and disjointed. It is not easy to understand how exactly the model is trained. I know space is limited, but even an algorithm in the supplement in addition to the loss function would help
>
> **A1:** Thank you for the suggestion, we will consolidate all training details into a single section and add a training algorithm to the Appendix in the updated draft.
>
> > **Q2:** One concern would be that such an optimization causes adverse effects on the fidelity not being trained.
>
> **A2:** The loss function we use while training considers molecules from all fidelity levels 1...K, regardless of the current level k. In other words, while k dictates which fidelity data we will add to the dataset, the entire dataset is used to train the model regardless of k. This ensures that training at level k does not degrade performance at the other fidelity levels. Analyzing the reconstruction accuracy, and the accuracy of the surrogate models, confirms that they retain their performance when other fidelities are being trained.
>
> > **Q3:** The usual generation quality measures such as validity, diversity, and synthesizability are not presented
>
> **A3:** Among the generated compounds for both targets following filtering, the diversity (1 - mean pairwise Tanimoto similarity) is 0.81 for BRD4(2) and 0.83 for c-MET. The synthesizability (mean SAscore) is 3.6 for BRD4(2) and 3.5 for c-MET. The drug-likeness (mean QED) is 0.59 for BRD4(2) and 0.63 for c-MET. The validity is 100% for both targets, because we use SELFIES strings that guarantee validity. These metrics are in the range of typical drug compounds, so we consider them satisfactory. We will include these numbers in the updated draft.
>
> > **Q4:** Since all the baselines don’t have statistically significant results, it is a bit concerning to compare the results
>
> **A4:** As reported in Section 4.3, compounds generated by MF-LAL had better ABFE scores than all baseline methods at a statistically significant level (see Appendix B.3 for statistical details). Specifically, the difference between both the mean ABFE scores of all generated compounds, as well as the scores of the top 3 compounds, is significant between MF-LAL and each baseline for both targets. For the “# of active scaffolds” test, MF-LAL also produced significantly more active scaffolds than other baselines. Since MF-LAL shows statistically significant improvements over baseline methods, we do not think these comparisons are concerning.
>
> > **Q5:** Substructure similarity is measured with Tanimoto distance of Morgan fingerprints which does not capture 3D information.
>
> **A5:** If the reviewer is referring to the similarity metric we used to compute the number of active scaffolds, we do not think that capturing 3D information is critical in this case. Scaffold similarity, which is what we measure in our paper, is the most commonly used diversity metric by medicinal chemists and strongly relates to the overall structural diversity of a set of compounds [1]. Additionally, measuring the 3D shape diversity is a difficult task, and is not commonly done by practitioners [1].
>
> [1] Galloway et al. “Diversity-oriented synthesis as a tool for the discovery of novel biologically active small molecules.” Nature Communications 2010.

---

### Decision · Program_Chairs · 2025-05-01

**Decision:**

Accept (poster)

**Comment:**

This is a nice paper, and all reviewers generally had positive comments about the novelty, appropriateness, and applicability of the proposed multi-fidelity approach. All reviewers agreed the problem setting is important and the methodology is sound. All reviewers had various more minor concerns, mostly regarding evaluation or baselines. The authors provided detailed rebuttals for most issues; the consensus across all reviewers was a "weak accept", and I think based on its merits the paper would be a good contribution to ICML.